# *DENOISER*: Rethinking the Robustness for Open-Vocabulary Action Recognition

## Abstract

As one of the fundamental video tasks in computer vision, Open-Vocabulary Action Recognition (OVAR) has recently gained increasing attention, with the development of vision-language pre-trainings. To enable open-vocabulary generalization, existing methods formulate vanilla OVAR to evaluate the embedding similarity between visual samples and text descriptions. However, one crucial issue is completely ignored: the text descriptions given by users may be noisy, *e.g.*, misspellings and typos, limiting the real-world practicality. To fill the research gap, this paper analyzes the noise rate/type in text descriptions by full statistics of manual spelling; then reveals the poor robustness of existing methods; and finally rethinks to study a practical task: noisy OVAR. One novel *DENOISER* framework, covering two parts: generation and discrimination, is further proposed for solution. Concretely, the generative part denoises noisy text descriptions via a decoding process, *i.e.*, proposes text candidates, then utilizes inter-modal and intra-modal information to vote for the best. At the discriminative part, we use vanilla OVAR models to assign visual samples to text descriptions, injecting more semantics. For optimization, we alternately iterate between generative-discriminative parts for progressive refinements. The denoised text descriptions help OVAR models classify visual samples more accurately; in return, assigned visual samples help better denoising. We carry out extensive experiments to show our superior robustness, and thorough ablations to dissect the effectiveness of each component.

## 1 Introduction

Action recognition is one of the fundamental tasks in computer vision that involves classifying videos into meaningful semantics. Despite huge progress that has been made, existing researches focus more on closed-set scenarios, where action classes remain constant during training and inference. Such scenarios are an oversimplification of real life, and thus limiting their practical application. Recently, another line of research considers one more challenging scenario, namely open-vocabulary action recognition (OVAR), and receives increasing attention.

OVAR allows users to give free texts to describe action classes, and the model needs to match novel (unseen) text descriptions to videos with similar semantics. To tackle OVAR task, Vision-Language Alignment (VLA) paradigm [41, 14, 57] provides one preliminary but popular idea, *i.e.*, measuring the embedding similarity between text descriptions and video embeddings. Following this paradigm, recent works focus on minor improvements, *e.g.*, better align vision-language modalities [16, 49, 62]. Although promising, these works all maintain one unrealistic assumption in real-world scenarios, *i.e.*, the given text descriptions are absolutely clean/accurate. The concrete form is that they evaluate open-vocabulary performance by re-partitioning closed-set datasets in which text descriptions of classes are fully human-checked. But in fact, under real-world OVAR, novel text descriptions provided by users are sometimes noisy. Character misspellings (typos, missing, tense error) are inevitable [43, 25] in

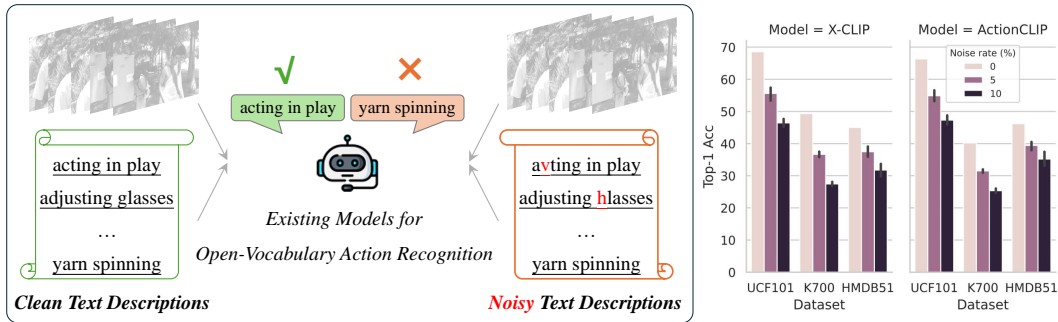

Figure 1: **Left**: For open-vocabulary action recognition (OVAR), existing researches neglect an essential aspect: the text descriptions provided by users may be noisy (*e.g.*, misspelling and typos), resulting in potential classification errors and limiting the real-world practicality. **Right:** Rethinking the robustness for popular OVAR methods [49, 62]. On various datasets, they exhibit high sensitivity to text noises. Besides, as the noise level increases, the performance degrades significantly.

thousands of descriptions, since users often don't double-check, as well as differences in user habits and diversity of scenarios (Fig. 1 Left).

We are hence motivated to fill the research gap of noisy text descriptions in OVAR. We analyze the noise rate/type in real-world corpora [26, 45, 3]. We also make comprehensive simulations of text noises, following NLP literature [42, 47]. Fig. 1 Right empirically evaluates noise hazards for existing OVAR methods [16, 49, 62]. One can find that just a small amount of noise lowers recognition accuracy by a large margin, implying quite poor robustness.

To spur the community to deal with the noisy OVAR task, being necessary and practical, this paper bravely faces the challenges. One vanilla idea is using a separate language model (*e.g.*, GPT [1]) to correct noisy class descriptions, and then adapt the off-the-shelf vision-language paradigm [41, 14, 57]. However, there exist two nettlesome issues. 1) *Textual Ambiguity*. One text description is usually a few compact words, with vague semantics, *e.g.*, for the noisy text "boird", there could be multiple cleaned candidates in terms of spelling, such as "bird" and "board". This short text lacks context, making phrase correction difficult for uni-modal language models. 2) *Cascaded Errors*. Text correction and action recognition are independently completed, without sharing knowledge. The noisy output of text correction is cascaded to the input of action recognition, resulting in continuous propagation of errors. To address these issues, we design one multi-modal robust framework: *DENOISER*.

Our first insight is to treat denoising of text descriptions as one *generative* task: given noisy text descriptions, decode the clean ones, by considering text-vision information to help denoising. Specifically, it consists of three components: text proposals, inter-modal weighting, and intra-modal weighting. We first propose potential text candidates based on spelling similarity to limit the decoding space. Then, two types of weighting are combined to decide the best candidate, that is, inter-modal weighting uses assigned visual samples to vote; while intra-modal weighting relies solely on text information. Our other insight is employing existing OVAR models as off-the-shelf tools to assign visual samples at *discriminative* step. Such tools have been proven to handle clean OVAR tasks well, also making our framework easier to adapt to previous models. For full usage of information in the same semantics, we then assign detail-rich visual samples to clarify the semantic ambiguity of compact text descriptions. To further avoid cascaded errors, we propose a solution of alternating iterations, to connect *generative* and *discriminative* steps. By progressive refinement, denoised text descriptions help OVAR models to match visual samples more accurately; assigned visual samples help better denoising. Under multiple iterations, denoising results and OVAR are both better.

Our main contributions are summarized as follows:

• We pioneer to explore noisy text descriptions for open-vocabulary action recognition (OVAR): first fully analyze the noise rate/type in text descriptions by extensive statistics in real-world corpora; then evaluate the robustness for existing methods; finally rethink to study one practical task: noisy OVAR.

• We propose a novel *DENOISER* framework to tackle the noisy OVAR task, by alternately optimizing generative-discriminative steps. The generative step leverages knowledge of vision-text alignment to denoises noisy text descriptions, in the form of progressive decoding; while the discriminative step assigns visual samples to text descriptions for open-vocabulary action recognition.

77 • We carry out extensive experiments to show the superior robustness of *DENOISER* against noisy
78 text descriptions, under various noises and datasets. Great performance improvements are achieved
79 over existing competitors. Thorough ablations are studied to show effectiveness of every design.

## 2 Related Work

81 **Vision-Language-Audio Pre-training** (VLP) aims to jointly optimize multi-modal embeddings with
82 large-scale web data, *e.g.*, CLIP [41], ALIGN [14], Florence [57], FILIP [55], VideoCLIP [52], and
83 LiT [58]. In architectures, VLP uses independent encoders for vision, text, and audio, followed by
84 cross-modal fusion. For optimization, contrastive learning [5, 61] and cross-modal matching [7, 29]
85 are mainstream, covering self supervision [32, 34], weak supervision [28, 8] and partial supervi-
86 sion [19, 33]. VLP benefits various applications: image-text retrieval [6, 18], video understand-
87 ing [23, 20, 22, 21], action recognition [16, 60], visual grounding [32, 56, 31], AIGC [4, 36].

88 **Open-Vocabulary Concept Learning** aims to understand vision, where conceptual semantics are
89 described by free/arbitrary text descriptions. It is characterized by using vision-language pre-trainings
90 to match text descriptions and visual samples in semantic space. Its typical evaluation metric is
91 the downstream zero-shot performance, *i.e.*, classify unseen classes [49, 62, 17, 38, 54, 48, 37]. To
92 achieve the evaluation, most methods re-partition closed-set datasets.[49] Although there is some
93 plausibility, such re-partition implicitly makes an unrealistic assumption: text descriptions of unseen
94 classes are human-checked, and thus absolutely clean, limiting real-world application. We pioneer
95 taking noises from text descriptions (misspellings and typos) into consideration. By adding real-world
96 noise for the above methods, we reveal their poor robustness, and design *DENOISER* for solution.

97 **Robustness of Language Models** is extensively studied by adversarial attack-defense techniques [50,
98 59]. When text inputs are facing noises, defense methods correct the outputs, dividing into: detection-
99 purification [63, 39], as well as adversarial training [53, 9, 35, 30, 51]. The former methods detect
100 and correct the corrupted part of a text phrase. The latter trains a model on adversarial samples to
101 increase its direct noise-against ability. Overall, all these methods employ solely textual information
102 for robustness in pure NLP tasks. We differ from them by considering robustness in the context of
103 multi-modal scenarios and by employing multi-modal information to better assist text denoising.

## 3 Method

105 We explore noisy text descriptions for open-vocabulary action recognition. In Sec 3.1, we introduce
106 noisy open-vocabulary setting; in Sec 3.2, we detail our *DENOISER* framework, covering *generative*
107 *- discriminative* sub-parts; in Sec 3.3, we report the accompanying optimization strategy.

### 3.1 Preliminary & Rethinking

109 **Open-Vocabulary Action Recognition (OVAR).** For a video dataset $\mathcal{V} = (v_j \in \mathbb{R}^{T \times H \times W \times 3})_j^N$,
110 OVAR aims to train one model $\Phi_{\text{OVAR}}$ that matches target videos with arbitrary text description $\mathcal{T}$.

$$\mathcal{Y}^{\text{train}} = \Phi_{\text{OVAR}}(\mathcal{V}^{\text{train}}, \mathcal{T}^{\text{train}}) \in \mathbb{R}^{C_{\text{base}}}, \quad \mathcal{Y}^{\text{test}} = \Phi_{\text{OVAR}}(\mathcal{V}^{\text{test}}, \mathcal{T}^{\text{test}}) \in \mathbb{R}^{C_{\text{novel}}}, \quad (1)$$

111 where $\mathcal{Y}$ refers to the matching label between $\mathcal{V}$ and $\mathcal{T}$. During training, (video, text, matching label)
112 triplets from the base semantic-classes are provided; while during testing, the model is evaluated
113 on the novel semantic-classes. Note that, the semantic-classes between training ($C_{\text{base}}$) and testing
114 ($C_{\text{novel}}$) are disjoint, *i.e.*, $C_{\text{base}} \cap C_{\text{novel}} = \varnothing$.

115 **Vision-Language Alignment (VLA).** To enable open-vocabulary capability, recent OVAR stud-
116 ies [16, 49, 62, 40] embrace vision-language pre-trainings (VLPs), for their notable ability in cross-
117 modal alignment. Specifically, OVAR could be achieved by measuring the embedding similarity
118 between text descriptions $\mathcal{T}$ and video samples $\mathcal{V}$, which is formally formulated as:

$$\mathcal{Y} = \sigma(\mathcal{F}_v * \mathcal{F}_t), \quad \mathcal{F}_v = \Phi_{\text{pool}}(\Phi_{\text{vis}}(\mathcal{V})) \in \mathbb{R}^{N \times D}, \quad \mathcal{F}_t = \Phi_{\text{txt}}(\mathcal{T}) \in \mathbb{R}^{C \times D}. \quad (2)$$

119 where $\sigma$ refers to the softmax activation, $\Phi_{\text{pool}}$ is the spatio-temporal pooling, $\Phi_{\text{vis}}$ and $\Phi_{\text{txt}}$ are
120 visual and textual encoders of VLPs, $D$ is the embedding dimension.

121 **Noisy Text Descriptions in OVAR.** Although great progress has been made, the VLA paradigm
122 suffers from an unrealistic assumption, *i.e.*, that text descriptions are absolutely clean/accurate,

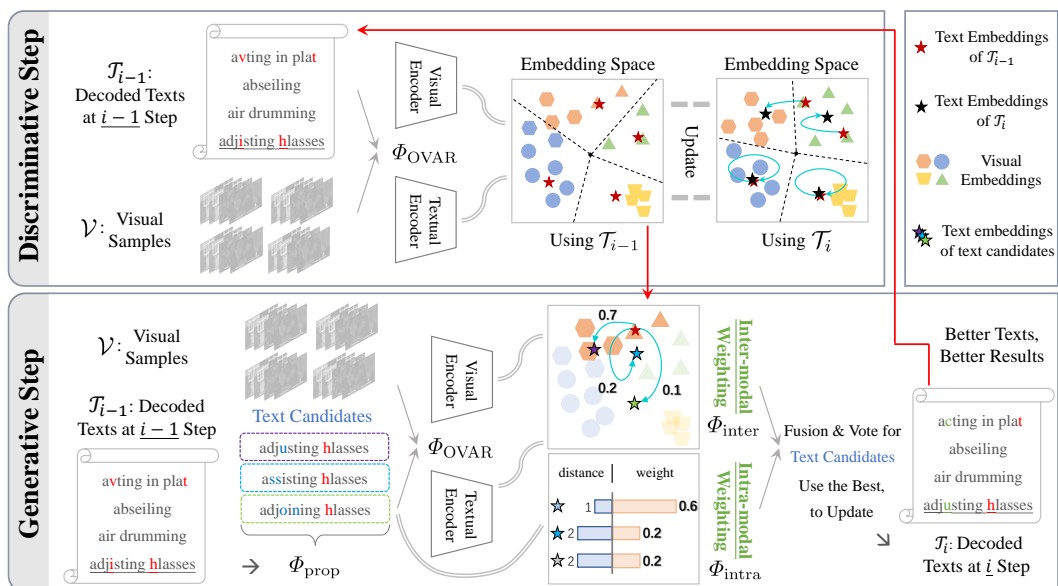

Figure 2: **Framework Overview**. *DENOISER* is composed of one *generative* part $\Psi_{\text{gene}}$ and one *discriminative* part $\Psi_{\text{disc}}$. $\Psi_{\text{gene}}$ views denoising text descriptions as a decoding process $\mathcal{T}_{i-1} \rightarrow \mathcal{T}_i$. We first propose text candidates $\Phi_{\text{prop}}$ for $\mathcal{T}_{i-1}$ based on spelling similarity; then choose the best candidate by inter-modal weighting $\Phi_{\text{inter}}$ and intra-modal weighting $\Phi_{\text{intra}}$. $\Phi_{\text{inter}}$ uses vision-text information, while $\Phi_{\text{intra}}$ relies solely on texts. $\Psi_{\text{disc}}$ assigns text semantics to visual samples, then only visual samples with the same semantics can vote for text candidates. We optimize alternatively between *generative* and *discriminative* steps to tackle noisy OVAR.

limiting the practicality in reality. Actually, the diversity of users and scenarios can easily cause text descriptions given to be somewhat noisy, especially for unseen semantic-classes, due to their enormous degree of freedom. Formally, for one text description with $n$ words, the clean/noisy versions $\mathcal{T}/\mathcal{T}'$ are:

$$\mathcal{T}' = (t'_1, \cdots, t'_n) = \Psi_{\text{noise}}(\mathcal{T}; p), \quad \mathcal{T} = (t_1, \cdots, t_n). \tag{3}$$

where $t_i$ is the $i$-th word of $\mathcal{T}$. $\Psi_{\text{noise}}$ refers to noise contamination in reality, *e.g.*, *inserting*, *substituting* and *deleting* characters with probability $p$, following [42, 47]. Since these three atomic operations defined in Levenshtein edit distance $\mathcal{D}$ are of distance 1, noise rate $p$ can also be deduced by:

$$p = \frac{\mathcal{D}(\mathcal{T}, \mathcal{T}')}{\max(\text{length of } \mathcal{T}, \text{length of } \mathcal{T}')} \tag{4}$$

As a result, the noisy OVAR task can be formulated as: given $\mathcal{V}$ and $\mathcal{T}'$, the model is expected to maximize the accuracy of action recognition, and even recovering $\mathcal{T}'$ to $\mathcal{T}$.

**Robustness of Existing Methods.** Fig. 1 evaluates for typical OVAR studies [49, 62], across three public datasets. In terms of Top-1 classification accuracy, existing methods are rather sensitive to noise and show one trend: the larger the noise, the more significant the performance degradation (please see quantitative experiments in Tab. 2). Such poor robustness to the noisy OVAR task, proves excessive idealization of existing studies and also motivates us to fill the research gap.

### 3.2 *DENOISER*: One Robust OVAR Framework

**Motivation.** Given the complexity of noisy OVAR, we here divide it into two sub-steps: denoising of text descriptions, and then vanilla OVAR. The former is viewed as one *generative* decoding form, by considering both vision-text information for progressive denoising. While the latter is in one natural *discriminative* form, by assigning text descriptions to video samples. For the joint optimization of these two sub-steps, we iterate alternately between *generative* and *discriminative* forms. As a result, our *DENOISER* framework progressively tackles the noisy OVAR task.

**Framework.** As shown in Fig. 2, our *DENOISER* framework covers two components: *generative* sub-step $\Psi_{\text{gene}}$ and *discriminative* sub-step $\Psi_{\text{disc}}$. For $\Psi_{\text{gene}}$, we iteratively refine text descriptions by one decoding process, that is, $(\mathcal{T}_0, \mathcal{T}_1, \cdots, \mathcal{T}_n)$, where $n$ is the index of decoding steps. Upon finishing step $i$, we will have $\mathcal{T}_i = (\overline{t_1}, \cdots, \overline{t_i}, t'_{i+1}, \cdots, t'_n)$, where $\overline{t}$ refers to the decoded version of $t$, meaning that the $i$-th word of text descriptions is decoded at step $i$. We start with $\mathcal{T}_0 = \mathcal{T}'$, and finish at $\mathcal{T}_n$ to ensure that all words are denoised. While for $\Psi_{\text{disc}}$, we find it identical to vanilla OVAR task and thus leveraging the VLA pipeline [16, 49] for help, which is off-the-shelf and well-studied. Formally, our *DENOISER* framework tackles noisy OVAR as follows:

$$\mathcal{T}_i = \Psi_{\text{gene}}(\mathcal{T}_{i-1}, \mathcal{Y}_{i-1}, \mathcal{V}), \quad \mathcal{Y}_{i-1} = \Psi_{\text{disc}}(\mathcal{T}_{i-1}, \mathcal{V}) = \Phi_{\text{OVAR}}(\mathcal{T}_{i-1}, \mathcal{V}). \tag{5}$$

At the *discriminative* step, we calculate the matching label $\mathcal{Y}_{i-1}$ to make coarse semantic classification of visual samples, *i.e.*, assign $\mathcal{T}_{i-1}$ to $\mathcal{V}$. At the *generative* step, we first propose $K$ text candidates $\Phi_{\text{prop}}(\mathcal{T}_{i-1})$ for $\mathcal{T}_i$ base on $\mathcal{T}_{i-1}$ to limit the decoding space. Then, to vote for the best candidate, we design two novel modules, namely inter-modal weighting $\Phi_{\text{inter}}$ and intra-modal weighting $\Phi_{\text{intra}}$. Here, $\Phi_{\text{inter}}$ uses vision information $\mathcal{V}$, while $\Phi_{\text{intra}}$ relies on text information $\mathcal{T}_{i-1}$.

We alternate between the *generative* and *discriminative* steps to optimize the decoding result step by step. Please find in Algorithm 1 for comprehensive details.

### 3.3   Optimization for the *DENOISER* Framework

**Discriminative Step** consists in calculating cross-modal matching labels $\mathcal{Y}$ using $\Psi_{\text{disc}}$. Intuitively, visual samples $\mathcal{V}_c$ whose labels $\mathcal{Y}$ are assigned to semantic-class $c$, *i.e.* $\operatorname{argmax} \mathcal{Y} = c$, are those who could help decode $\mathcal{T}_{c,i}$ most efficiently. On the contrary, visual samples from other semantic-classes may have few connections with the current class and thus provide no meaningful aid. Here, we find this process is identical to vanilla OVAR, and hence employs $\Phi_{\text{OVAR}}$ as $\Psi_{\text{disc}}$. We theoretically prove in the Appendix that, $\mathcal{V}_c$ is the best set of visual samples to choose from. With $\mathcal{V}_c$ defined and $\operatorname{argmax} \mathcal{Y} = c$, $\Psi_{\text{gene}}$ decodes text descriptions $\mathcal{T}_{c,i}$ for each semantic-class $c$:

$$\Psi_{\text{gene}}(\mathcal{T}_{c,i-1}, \mathcal{Y}, \mathcal{V}) = \Psi_{\text{gene}}(\mathcal{T}_{c,i-1}, \mathcal{V}_c) = \operatorname*{argmax}_{\mathcal{T}_{c,i}} p(\mathcal{T}_{c,i} | \mathcal{T}_{c,i-1}, \mathcal{V}_c). \tag{6}$$

Recall $t_{c,i}$ is the $i$-th word to be decoded, and $\mathcal{T}_{c,i-1}$ is from last decoding, with the first $i-1$ words decoded. As we decode word-by-word, choosing the best $\mathcal{T}_{c,i}$ is exactly choosing the best $t_{c,i}$, *i.e.* $\operatorname{argmax}_{\mathcal{T}_{c,i}} p(\mathcal{T}_{c,i} | \mathcal{T}_{c,i-1}, \mathcal{V}_c) = \operatorname{argmax}_{t_{c,i}} p(t_{c,i} | \mathcal{T}_{c,i-1}, \mathcal{V}_c)$, as we do in *generative* step.

**Generative Step** here consists in, for each semantic-class $c$, choosing the best $t_{c,i}$ that maximizes $p(t_{c,i} | \mathcal{T}_{c,i-1}, \mathcal{V}_c)$. With $p(\mathcal{T}_{c,i-1}, \mathcal{V}_c)$ and $p(\mathcal{V}_c)$ same for all possible $t_{c,i}$, we make detailed derivations in the Appendix to show that:

$$p(t_{c,i} | \mathcal{T}_{c,i-1}, \mathcal{V}_c) \propto p(t_{c,i}, \mathcal{T}_{c,i-1}, \mathcal{V}_c) \propto \prod_{v_j \in \mathcal{V}_c} p(t_{c,i} | v_j) p(\mathcal{T}_{c,i-1} | t_{c,i}, v_j). \tag{7}$$

Here, the error model $p(\mathcal{T}_{c,i-1} | t_{c,i}, v_j)$ evaluates how $t_{c,i}$ may be misspelled as $t'_{c,i}$, since the $i$-th word in $\mathcal{T}_{c,i-1}$ is still noisy and not decoded. Knowing that errors in text descriptions are independent of visual samples, it reduces to uni-modal $p(\mathcal{T}_{c,i-1} | t_{c,i})$. As the error that one may make given the correct text is harder to model while the reverse is much easier, we let $p(\mathcal{T}_{c,i-1} | t_{c,i}) \propto p(t_{c,i} | \mathcal{T}_{c,i-1})$. Please refer to detailed derivations in the Appendix. As a result, our final objective is:

$$p(t_{c,i} | \mathcal{T}_{c,i-1}) \prod_{v_j \in \mathcal{V}_c} p(t_{c,i} | v_j) = \Phi_{\text{intra}} \prod_{v_j \in \mathcal{V}_c} \Phi_{\text{inter}}. \tag{8}$$

*Text Proposals* consists in proposing $K$ candidates $\{t_i^k\}_k$ for $t_i$ with the lowest Levenshtein Edit Distance $\mathcal{D}(\cdot, t'_i)$ (a metric of spelling similarity). By replacing original noisy word $t'_i$ in $\mathcal{T}_{i-1}^k$ with $\{t_i^k\}_k$, they form $\Phi_{\text{prop}}(\mathcal{T}_{i-1}) = \mathcal{T}_i^k = (\overline{t_1}, \cdots, \overline{t_{i-1}}, t_i^k, t'_{i+1}, \cdots, t'_n)$, the $K$ candidates for $\mathcal{T}_i$. The benefit of text proposals is to reduce computing complexity. Since text embeddings are quantized in the semantic space, the search is limited to proposed candidates, rather than in the entire space.

*Inter-modal Weighting* $\Phi_{\text{inter}} = p(t_{c,i} | v_j)$, $v_j \in \mathcal{V}_c$ relies on vision samples from semantic-class $c$ to determine the best $t_{c,i}$ for the next iteration. Concretely, we model the probability of being chosen

**Algorithm 1** *DENOISER*: Robust Open-Vocabulary Action Recognition

---

**Require:** noisy text descriptions $\mathcal{T}'$, visual samples $\mathcal{V}$, iteration number $n$, temperature $\lambda$, candidate number $K$, edit distance $\mathcal{D}$, open-vocabulary model $\Phi_{\text{OVAR}}$

$\mathcal{T}_0 \leftarrow \mathcal{T}'$

**for** $i = 1, 2, \cdots, n$ **do**

    **for** $c = 1, 2, \cdots, C$ **do**                                             ▷ Text Proposals

        $t'_{c,i}$ is the $i$-th word of $\mathcal{T}_{c,i-1}$, which is noisy and not yet decoded

        Select from corpus, $K$ candidates $\{t^k_{c,i}\}_k$ with the smallest $\mathcal{D}$ with $t'_{c,i}$

        Replace $t'_{c,i}$ with $\{t^k_{c,i}\}_k$, forming $\{\mathcal{T}^k_{c,i}\}_k$

    **end for**

    **for** $j = 1, 2, \cdots, |\mathcal{V}|$ **do**                                  ▷ Discriminative Step

        $c \leftarrow \underset{c}{\arg\max} \underset{k}{\max} \dfrac{\exp(\mathcal{S}(v_j, \mathcal{T}^k_{c,i}))}{\sum_{k'} \exp(\mathcal{S}(v_j, \mathcal{T}^{k'}_{c,i}))}$

        Assign $v_j$ to class $c$, $v_j \in \mathcal{V}_c$

    **end for**

    **for** $c = 1, 2, \cdots, C$ **do**                                        ▷ Generative Step

        $\Phi^k_{\text{intra}} \leftarrow \dfrac{\exp(-\mathcal{D}(t^k_{c,i}, t'_{c,i})/\lambda)}{\sum_{k'} \exp(-\mathcal{D}(t^{k'}_{c,i}, t'_{c,i})/\lambda)}$         ▷ Intra-Modal Weighting

        $\Phi^k_{\text{inter}} \leftarrow \prod_{v_j \in \mathcal{V}_c} \dfrac{\exp(\mathcal{S}(v_j, \mathcal{T}^k_{c,i}))}{\sum_{k'} \exp(\mathcal{S}(v_j, \mathcal{T}^{k'}_{c,i}))}$         ▷ Inter-Modal Weighting

        $\mathcal{T}_{c,i} \leftarrow \mathcal{T}^k_{c,i}$, $k = \arg\max_k \Phi^k_{\text{intra}} \times \Phi^k_{\text{inter}}$

    **end for**

**end for**

---

for each proposed candidate to be:

$$\mathbb{P}(t_{c,i} = t^k_{c,i} | v_j) = \mathbb{P}(\mathcal{T}_{c,i} = \mathcal{T}^k_{c,i} | v_j) = \frac{\exp(\mathcal{S}(v_j, \mathcal{T}^k_{c,i}))}{\sum_{k'} \exp(\mathcal{S}(v_j, \mathcal{T}^{k'}_{c,i}))}, \ v_j \in \mathcal{V}_c. \tag{9}$$

where $\mathcal{S}(\cdot, \cdot)$ is the cosine similarity between video-text embeddings, both encoded by $\Phi_{\text{OVAR}}$. The intuition is that the more unanimously visual samples agree on candidate $\mathcal{T}^k_{c,i}$, the more likely it is the text descriptions corresponding to semantic-class $c$. Besides, by letting visual samples vote on $\mathcal{T}^k_{c,i}$ instead of $t^k_{c,i}$, we take into consideration not only the current word $t_{c,i}$ but also context implicitly.

*Intra-modal Weighting* $\Phi_{\text{intra}} = p(t_{c,i} | \mathcal{T}_{c,i-1})$ relies solely on text information to decide the best $t_{c,i}$ for next iteration. Although $\Phi_{\text{intra}}$ may be solved by uni-modal spell-checkers [15] or large language models [1], we here design a simple model by considering only spelling similarity (ignore contexts), to save computing costs. That is, choose $t_{c,i}$ depending solely on $t'_{c,i}$ instead of on entire $\mathcal{T}_{c,i-1}$:

$$\mathbb{P}(t_{c,i} = t^k_{c,i} | \mathcal{T}_{c,i-1}) = \mathbb{P}(t_{c,i} = t^k_{c,i} | t'_{c,i}) = \frac{\exp(-\mathcal{D}(t^k_{c,i}, t'_{c,i})/\lambda)}{\sum_{k'} \exp(-\mathcal{D}(t^{k'}_{c,i}, t'_{c,i})/\lambda)}. \tag{10}$$

The intuition is that, the more similar a word candidate $t^k_{c,i}$ is, compared to the noisy word $t'_{c,i}$, the more likely it is the corresponding denoised word. Here, we introduce one temperature parameter $\lambda$ to balance $\Phi_{\text{intra}}$ and $\Phi_{\text{inter}}$. A larger $\lambda$ indicates that different edit distance gives similar probabilities, meaning that we rely more on visual samples for decision, and vice versa.

## 4   Experiments

**Typical Models for Vanilla OVAR**. To illustrate the generalizability of our framework, we leverage two typical models from the VLA pipeline as $\Phi_{\text{OVAR}}$, that is, ActionCLIP [49] and XCLIP [62]. These two models adopt hand-crafted prompts and visual-conditioned prompt tuning, respectively. Under both models, we choose ViT-B/16-32F as the network backbones, for simplicity.

**Datasets**. HMDB51 [26] contains 7k videos covering 51 action categories. UCF101 [45] contains 13k videos spanning 101 action categories. Kinetics700 [3] (K700) is simply an extension of K400, with around 650k video clips sourced from YouTube. To partition these datasets for open-vocabulary action recognition, this paper follows the standard consensus [49, 62], for the sake of fairness.

Figure 3: **Statistics for Noises in Reality**. Text noises may be classified into 4 types: inserting, substituting, swapping, and deleting characters.[2] In terms of edit distance, based on TOEFL-Spell dataset[10], most of the text noises have an edit distance = 1 compared to the clean version. Nevertheless, the distribution tends to be positively skewed towards larger noise.

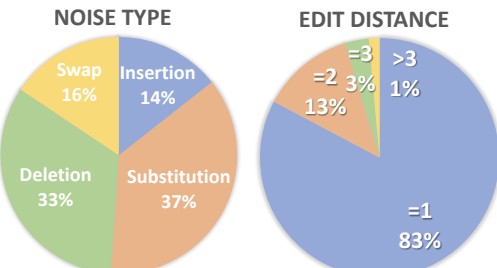

Table 1: **Comparisons between Various Competitors.** Using ActionCLIP [49] as $\Phi_{OVAR}$ while evaluating on UCF101, we compare with statistical text spell-checkers (PySpellChecker [15]), neural based ones (Bert from NeuSpell) [13], and GPT 3.5 [1]. Our method remarkably outperforms others in terms of Top-1 classification accuracy, and semantic similarity of recovered text descriptions.

| Noise Type | Noise Rate | Competitors | Top-1 Acc | Label Acc | Semantic Similarity |
|---|---|---|---|---|---|
| – | 0% | Upper Bound | 66.3 | 100 | 100 |
| Real | ~5.52% | GPT 3.5 [1] | $61.2_{\pm1.4}$ | $74.7_{\pm1.9}$ | $97.1_{\pm0.4}$ |
| | | Bert (NeuSpell) [13] | $56.0_{\pm1.1}$ | $64.7_{\pm2.0}$ | $94.5_{\pm0.4}$ |
| | | PySpellChecker [15] | $59.9_{\pm1.2}$ | $79.6_{\pm1.6}$ | $96.7_{\pm0.3}$ |
| | | **Ours** | $\mathbf{61.5_{\pm0.7}}$ | $\mathbf{82.3_{\pm1.6}}$ | $\mathbf{97.2_{\pm0.3}}$ |
| Simulated | 5% | GPT 3.5 [1] | $59.7_{\pm1.2}$ | $47.6_{\pm3.1}$ | $95.9_{\pm0.4}$ |
| | | Bert (NeuSpell) [13] | $56.6_{\pm0.5}$ | $66.2_{\pm2.3}$ | $94.6_{\pm0.4}$ |
| | | PySpellChecker [15] | $60.9_{\pm1.1}$ | $82.5_{\pm2.9}$ | $97.1_{\pm0.4}$ |
| | | **Ours** | $\mathbf{63.8_{\pm0.7}}$ | $\mathbf{86.4_{\pm2.3}}$ | $\mathbf{97.7_{\pm0.2}}$ |
| | 10% | GPT 3.5 [1] | $58.5_{\pm1.3}$ | $51.6_{\pm2.3}$ | $95.8_{\pm0.3}$ |
| | | Bert (NeuSpell) [13] | $51.0_{\pm0.5}$ | $50.4_{\pm3.6}$ | $91.6_{\pm0.6}$ |
| | | PySpellChecker [15] | $55.7_{\pm1.1}$ | $69.3_{\pm1.5}$ | $94.8_{\pm0.3}$ |
| | | **Ours** | $\mathbf{61.2_{\pm0.8}}$ | $\mathbf{75.9_{\pm1.9}}$ | $\mathbf{96.4_{\pm0.3}}$ |

**Metric.** We use three metrics for full evaluations from multiple perspectives. Top-1 Acc refers to the top-1 classification accuracy of noisy open-vocabulary action recognition. Label Acc counts the percentage of denoised text descriptions that match exactly with ground truth. Semantic Similarity calculates the cosine similarity of embeddings, between denoised and clean text descriptions. Label Acc and Semantic Similarity measure how well noisy text descriptions are recovered.

**Implementations.** We set the proposal number $K = 10$. Intra-modal weighting and inter-modal weighting are both used to determine the best candidate. Temperature $\lambda$ follows a linear schedule from 0.01 to 1. We use the same corpus as in PySpellChecker, which contains 70317 English words, for text proposals. For typical OVAR methods [49, 62], we choose the ViT-B/16-32F checkpoint pretrained on K400 [24] to evaluate their zero-shot robustness on HMDB51 [27], UCF101 [46] and K700 [44]. Since K700 and K400 have overlapped categories, we exclude them when evaluating on K700. For UCF101, we use the separated lowercase text label. All ablation studies are conducted on UCF101 under 20% noise. For statistical significance, We do each simulation 10 times and report the mean and confidence interval of 95%. All experiments are done using a single RTX 3090.

### 4.1 Statistics on Noise Type/Rate for Text Descriptions

**Real Noise.** We adopt two large-scale corpora [11, 10] of misspellings to analyze noise type in text descriptions. As shown in Fig. 3, the conclusion is similar to the NLP community [42, 47], *i.e.*, three atomic types of noise are inserting, substituting, and deleting text characters. More complicated noise patterns, *e.g.* swaping, can be constructed by mixing atomic noise types. Then, following previous literature, we quantify noise rate through Levenshtein Edit Distance, a generally accepted metric, to calculate the occurrence number of atomic noise types. Specifically, GitHub Typo Corpus [11] contains over 350k edits of typos from GitHub. The average noise rate (per sentence) is 3.3%. Nevertheless, the distribution is highly positively skewed (skewness = 2.9). For the worst 5% cases, the noise rate (per sentence) is larger than 9.4%. TOEFL-Spell Corpus [10] samples essays written by candidates from various language backgrounds in TOEFL® iBT test. There are, on average, 6.9 spelling mistakes per essay. For misspelled words, the noise rate (per word) is on average 16.0%.

Table 2: **Comparison Across Datasets and Models**. On three standard datasets, facing multiple noise types (real or simulated), and under various noise rates, our *DENOISER* consistently improves the performance for noisy OVAR, regardless of underlying OVAR methods $\Phi_{\text{OVAR}}$.

| Dataset | Noise Type | Noise Rate | $\Phi_{\text{OVAR}}$: Typical Models for Vanilla OVAR task | | | |
|---|---|---|---|---|---|---|
| | | | ActionCLIP [49] | | XCLIP [62] | |
| | | | w/o Ours | **w Ours** | w/o Ours | **w Ours** |
| UCF101 | Upper Bound | | 66.3 | | 68.6 | |
| | Real | $\sim$5.52% | $54.0_{\pm 2.3}$ | $\mathbf{61.5_{\pm 0.7}}$ | $53.8_{\pm 2.7}$ | $\mathbf{63.4_{\pm 0.9}}$ |
| | Simulated | 5% | $54.9_{\pm 1.8}$ | $\mathbf{63.2_{\pm 0.7}}$ | $55.6_{\pm 2.2}$ | $\mathbf{64.2_{\pm 1.4}}$ |
| | | 10% | $47.3_{\pm 1.4}$ | $\mathbf{61.2_{\pm 1.2}}$ | $46.4_{\pm 1.3}$ | $\mathbf{62.9_{\pm 2.3}}$ |
| HMDB51 | Upper Bound | | 46.2 | | 45.0 | |
| | Real | $\sim$6.71% | $37.6_{\pm 1.6}$ | $\mathbf{40.0_{\pm 1.4}}$ | $35.3_{\pm 1.5}$ | $\mathbf{38.4_{\pm 1.4}}$ |
| | Simulated | 5% | $39.4_{\pm 1.4}$ | $\mathbf{41.3_{\pm 1.4}}$ | $37.5_{\pm 1.8}$ | $\mathbf{39.7_{\pm 1.0}}$ |
| | | 10% | $35.2_{\pm 2.3}$ | $\mathbf{39.6_{\pm 1.4}}$ | $31.8_{\pm 2.2}$ | $\mathbf{37.3_{\pm 1.5}}$ |
| K700 | Upper Bound | | 40.2 | | 49.3 | |
| | Real | $\sim$5.47% | $30.8_{\pm 0.51}$ | $\mathbf{35.9_{\pm 0.4}}$ | $35.6_{\pm 0.6}$ | $\mathbf{43.5_{\pm 0.7}}$ |
| | Simulated | 5% | $31.5_{\pm 0.5}$ | $\mathbf{36.8_{\pm 0.3}}$ | $36.7_{\pm 0.9}$ | $\mathbf{44.1_{\pm 0.6}}$ |
| | | 10% | $25.4_{\pm 0.8}$ | $\mathbf{35.3_{\pm 0.5}}$ | $27.5_{\pm 0.7}$ | $\mathbf{41.8_{\pm 0.9}}$ |

**Noise Scenarios.** In the "Simulated" noise type, we mix three atomic noises: insertion, substitution, and deletion. Concretely, for each character, we perturb it with probability $p$. For each perturbation, it will be insertion, substitution, and deletion with equal probability. To further ensure real-world generalizability, we ask GPT3.5 to give examples of perturbation according to real-world scenarios. We mix them into simulated noises. Noise rate $p$ of the "Real" noise type is estimated with Eq. (3).

## 4.2 Comparison with State-of-the-art Methods

**Comparison to Competitors.** Tab. 1 compares from three axes: Top-1 Acc of $\Phi_{\text{OVAR}}$ after correction, Label Acc and Semantic Similarity. PySpellChecker is a uni-modal statistical model that corrects each word by edit distance and appearance frequency. Bert (NeuSpell) [13] employs a uni-modal Bert-based model to translate noisy text descriptions into clean ones. We also ask GPT 3.5 to denoise text descriptions using the prompt "The following words may contain spelling errors by deleting, inserting, and substituting letters. You are a corrector of spelling errors. Give only the answer without explication. What is the correct spelling of the action of <noisy text description>?". Our method outperforms all competitors by large margins, which is impressive because our method is unsupervised without prior knowledge other than those contained in the OVAR model. Note that the output of GPT 3.5 tends to be unstable depending on prompts, which requires manual cleaning to remove irrelevant parts contained in the output, thus impeding real-world usage.

**Comparisons Across Datasets/Models.** Tab. 2 compares Top-1 Acc to further reveal our solution is scalable/generalizable. Under various noise rates, our model is robust to achieve huge improvements. In terms of scalability across models, our method is not only applicable to hand-crafted prompts as in ActionCLIP but also to learnable visual-conditioned prompts as in XCLIP. Furthermore, we notice that, whenever XCLIP outperforms ActionCLIP, our method also yields a better result. A better visual encoder and well-tuned prompt may significantly increase our performance, showing that our method's upper limit could become higher, as the community continues to train better OVAR models.

## 4.3 Ablation Study

**Inter-modal Weighting $\Phi_{\text{inter}}$ & Intra-modal Weighting $\Phi_{\text{intra}}$.** Tab. 3 shows that, both $\Phi_{\text{inter}}$ and $\Phi_{\text{intra}}$ contribute to denoising text descriptions and to improving the robustness of underlying $\Phi_{\text{OVAR}}$. In terms of Top-1 Acc and Semantic Similarity, $\Phi_{\text{inter}}$ performs better than $\Phi_{\text{intra}}$, since $\Phi_{\text{inter}}$ uses visual information as one direct optimization guideline to improve video recognition. While $\Phi_{\text{intra}}$ performs better in terms of Label Acc, which focuses more on spelling correctness. Besides, $\Phi_{\text{inter}}$ and $\Phi_{\text{intra}}$ turn out to be complementary: visual information helps to understand noisy text descriptions; while textual information prevents the model from being misled by visual samples. We achieve the best performance when combining these two weightings.

Table 3: **Ablations for Inter-modal Weighting** $\Phi_{\text{Inter}}$**, Intra-modal Weighting** $\Phi_{\text{Inter}}$**, Schedule of Temperature** $\lambda$. $\Phi_{\text{Inter}}$ alone outperforms $\Phi_{\text{Intra}}$. Both contribute to correcting class texts, and give the best results when combined. Linear schedule of balancing factor $\lambda$ outperforms the constant one, meaning that it helps to rely more on $\Phi_{\text{Intra}}$ at first, and then gradually switch to $\Phi_{\text{Inter}}$.

|  | $\Phi_{\text{Inter}}$ | $\Phi_{\text{Intra}}$ | Schedule $\lambda$ | Top-1 Acc | Label Acc | Semantic Similarity |
|---|---|---|---|---|---|---|
| A1 |  | ✓ | / | $48.1_{\pm 2.2}$ | $38.2_{\pm 2.5}$ | $88.9_{\pm 0.4}$ |
| A2 | ✓ |  | / | $52.9_{\pm 1.4}$ | $34.1_{\pm 2.4}$ | $89.1_{\pm 0.6}$ |
| A3 | ✓ | ✓ | Constant | $54.5_{\pm 2.5}$ | $54.9_{\pm 4.5}$ | $92.4_{\pm 0.8}$ |
| A4 | ✓ | ✓ | Linear | $\mathbf{55.2_{\pm 1.5}}$ | $\mathbf{55.1_{\pm 3.0}}$ | $\mathbf{92.9_{\pm 0.6}}$ |

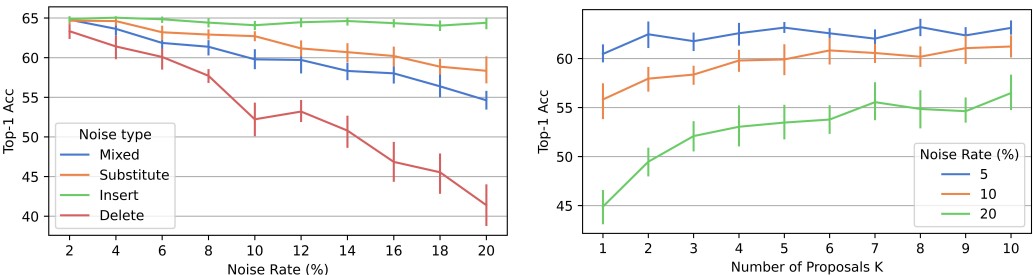

Figure 4: We evaluate on UCF101 by using ActionCLIP as $\Phi_{\text{OVAR}}$. **Left: Ablation Study on Noise Type**. "Mixed" means that all types of text noises: "Substitute", "Insert", "Delete" take place with equal probability. Our *DENOISER* shows good resilience, especially against noises of inserting or substituting. **Right: Ablation Study on Proposal Number** $K$. As $K$ increases, Top-1 Acc increases and converges gradually towards the upper bound, but it also brings heavier computing costs.

**Temperature Schedule** $\lambda$ balances intra-modal weighting and inter-modal weighting. One larger $\lambda$ indicates more reliance on inter-modal weighting. "Linear" means that $\lambda$ augments from 0.01 to 1 linearly. Tab. 3 reports that it is beneficial to rely more on intra-modal at the beginning of decoding, and then gradually turn to inter-modal for more help. This indicates that, when text noises are high, $\Phi_{\text{intra}}$ offers more help; when text noises are slight, $\Phi_{\text{inter}}$ could help more.

**Noise Type.** Fig. 4 Left reports our robustness under various noise types/rates. "Mixed" means that three noise types: "Substitute", "Insert", "Delete" are equally possible to appear. Our method shows remarkable resilience when texts are perturbed by inserting or substituting characters. Performance degradation is observed when texts are perturbed by deleting characters. It is reasonable, as deleting characters causes huge information loss, making the model difficult to recover clean text descriptions.

**Number of Candidates** $K$**.** Fig. 4 Right shows as $K$ increases, inter-modal weighting can reveal its full power, hence improving performance. Otherwise, if a good candidate is excluded from the proposal stage due to a small $K$, it can be selected by neither of the inter- or intra-modal weighting, thus decreasing performance. Moreover, the performance tends towards one plateau, showing a decreasing marginal contribution of more proposals to performance. Since a larger $K$ means more computing costs for text encoding, we select $K = 10$ by default to make reasonable trade-offs.

## 5 Conclusion

This paper investigates how noises in class-text descriptions negatively interference OVAR; and one novel framework *DENOISER* is proposed for solutions. By incorporating visual information during denoising, we clarify the ambiguity induced by short and context-lacking text descriptions; by iteratively refining the denoised output through one generative-discriminative process, we mitigate cascaded errors which may propagate from spell-checking models to outputs of OVAR model. We conduct extensive experiments to demonstrate the generalizability of *DENOISER* across multiple models and datasets, and also show our superiority over uni-modal spell-checking solutions.

**Limitations.** 1) We focus more on spelling noises; while in the real world, text noises can be more complex, involving semantic ambiguity. Equipping *DENOISER* with large language models may be a feasible solution. 2) Using more text candidates or visual samples brings better results for *DENOISER*, but also costs more. There is a trade-off between performance and computational cost.

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

# A Theoretical Analysis

## A.1 Decoding Objective

At each step $i$, the decoding objective to find $\operatorname{argmax}_{t_i} p(t_i|\mathcal{T}_{i-1}, \mathcal{V})$. Note that, $p(\mathcal{T}_{i-1}, \mathcal{V})$ is same

for all possible $t_i$. As a result, our objective is written as:

$$\operatorname*{argmax}_{t_i} p(t_i|\mathcal{T}_{i-1}, \mathcal{V}) = \operatorname*{argmax}_{t_i} p(t_i|\mathcal{T}_{i-1}, \mathcal{V})p(\mathcal{T}_{i-1}, \mathcal{V}) \tag{11}$$

$$= \operatorname*{argmax}_{t_i} p(t_i, \mathcal{T}_{i-1}, \mathcal{V}) \tag{12}$$

$$= \operatorname*{argmax}_{t_i} \log p(t_i, \mathcal{T}_{i-1}, \mathcal{V}) \tag{13}$$

## A.2 Discriminative Step

At the discriminative step, we choose the best set of $\mathcal{V}$ that helps decode $t_{c,i}$ for each semantic-class

$c$. To understand why $\mathcal{V}_c$, the set of visual samples $v_j$ whose labels $\mathcal{Y}_j$ are assigned to semantic-class

$c$ are those who help decode most efficiently, we first introduce a hidden discrete random variable

$z_j \sim Q_j$ for each $v_j$, indicating the index of class assignment. $z_j = c$ means that $\operatorname{argmax} \mathcal{Y}_j = c$.

Knowing that all visual samples are independent and using Jensen inequality:

$$\log p(t_i, \mathcal{T}_{i-1}, \mathcal{V}) = \sum_j \log p(t_i, \mathcal{T}_{i-1}, v_j) \tag{14}$$

$$= \sum_j \log \sum_{z_j} p(t_i, \mathcal{T}_{i-1}, v_j, z_j) \tag{15}$$

$$= \sum_j \log \sum_{z_j} Q_j(z_j) \frac{p(t_i, \mathcal{T}_{i-1}, v_j, z_j)}{Q_j(z_j)} \tag{16}$$

$$\geq \sum_j \sum_{z_j} Q_j(z_j) \log \frac{p(t_i, \mathcal{T}_{i-1}, v_j, z_j)}{Q_j(z_j)} \tag{17}$$

Equality is attained at $Q_j(z_j) \propto p(t_i, \mathcal{T}_{i-1}, v_j, z_j)$. Since $\sum_{z_j} Q_j(z_j) = 1$, to maximize the lower

bound, we have:

$$Q_j(z_j) = \frac{p(t_i, \mathcal{T}_{i-1}, v_j, z_j)}{\sum_{z_j} p(t_i, \mathcal{T}_{i-1}, v_j, z_j)} \tag{18}$$

$$= \frac{p(t_i, \mathcal{T}_{i-1}, v_j, z_j)}{p(t_i, \mathcal{T}_{i-1}, v_j)} \tag{19}$$

$$= p(z_j|t_i, \mathcal{T}_{i-1}, v_j) \tag{20}$$

$$= p(z_j|\mathcal{T}_i, v_j) \tag{21}$$

Given class texts and visual samples, the best estimation is:

$$\mathbb{P}(z_j = c|\mathcal{T}_i, v_j) = \begin{cases} 1 & c = \operatorname*{argmax}_c \max_k \frac{\exp(\mathcal{S}(v_j, \mathcal{T}_{c,i}^k))}{\sum_{k'} \exp(\mathcal{S}(v_j, \mathcal{T}_{c,i}^{k'}))} \\ 0 & \text{otherwise} \end{cases} \tag{22}$$

Note that, $Q_j$ is well defined because:

$$\lim_{Q_j(z_j) \to 0^+} Q_j(z_j) \log \frac{p(t_i, \mathcal{T}_{i-1}, v_j, z_j)}{Q_j(z_j)} = 0 \tag{23}$$

With $Q_j$ defined in this way, we find the discriminative step to be identical to how $\Phi_{\text{OVAR}}$ assigns labels. We have $Q_j(c) = 1$ only for $\{j|v_j \in \mathcal{V}_c\}$:

$$\log p(t_i, \mathcal{T}_{i-1}, \mathcal{V}) \geq \sum_j \sum_{z_j} Q_j(z_j) \log \frac{p(t_i, \mathcal{T}_{i-1}, v_j, z_j)}{Q_j(z_j)} \tag{24}$$

$$= \sum_c \sum_{j, v_j \in \mathcal{V}_c} \sum_{z_j} Q_j(z_j) \log \frac{p(t_i, \mathcal{T}_{i-1}, v_j, z_j)}{Q_j(z_j)} \tag{25}$$

$$= \sum_c \sum_{j, v_j \in \mathcal{V}_c} \log p(t_i, \mathcal{T}_{i-1}, v_j, z_j = c) \tag{26}$$

$$= \sum_c \log p(t_{c,i}, \mathcal{T}_{c,i-1}, \mathcal{V}_c) \tag{27}$$

$$\tag{28}$$

## A.3 Generative Step

We optimize $t_{c,i}$ for each semantic-class:

$$\operatorname*{argmax}_{t_{c,i}} \log p(t_{c,i}, \mathcal{T}_{c,i-1}, \mathcal{V}_c) = \operatorname*{argmax}_{t_{c,i}} p(t_{c,i}, \mathcal{T}_{c,i-1}, \mathcal{V}_c) \tag{29}$$

$$= \operatorname*{argmax}_{t_{c,i}} \prod_{v_j \in \mathcal{V}_c} p(t_{c,i}, \mathcal{T}_{c,i-1}, v_j) \tag{30}$$

$$= \operatorname*{argmax}_{t_{c,i}} \prod_{v_j \in \mathcal{V}_c} p(\mathcal{T}_{c,i-1}|t_{c,i}, v_j) p(t_{c,i}|v_j) p(v_j) \tag{31}$$

$$= \operatorname*{argmax}_{t_{c,i}} \prod_{v_j \in \mathcal{V}_c} p(\mathcal{T}_{c,i-1}|t_{c,i}, v_j) p(t_{c,i}|v_j) \tag{32}$$

Noting that $p(\mathcal{T}_{c,i-1})$ is the same for any possible $t_{c,i}$:

$$\operatorname*{argmax}_{t_{c,i}} p(\mathcal{T}_{c,i-1}|t_{c,i}, v_j) = \operatorname*{argmax}_{t_{c,i}} p(\mathcal{T}_{c,i-1}|t_{c,i}) \tag{33}$$

$$= \operatorname*{argmax}_{t_{c,i}} \frac{p(t_{c,i}|\mathcal{T}_{c,i-1}) p(\mathcal{T}_{c,i-1})}{p(t_{c,i})} \tag{34}$$

$$= \operatorname*{argmax}_{t_{c,i}} \frac{p(t_{c,i}|\mathcal{T}_{c,i-1})}{p(t_{c,i})} \tag{35}$$

It is possible to optimize with prior $p(t_{c,i})$ by considering that the more a word is frequent, the less it is likely to be misspelled in real-world scenarios. In this paper, for simplicity, we assume the $t_{c,i}$ to be uniform:

$$\operatorname*{argmax}_{t_{c,i}} p(\mathcal{T}_{c,i-1}|t_{c,i}, v_j) = \operatorname*{argmax}_{t_{c,i}} p(t_{c,i}|\mathcal{T}_{c,i-1}) \tag{36}$$

# B  Additional Experiments

## B.1  DENOISER *vs*. Adversarial Training

Fig. 5 studies how adversarial training might mitigate the noise in text descriptions. We first train ActionCLIP ViT-B/32-8F from scratch on K400 by randomly injecting noise in its text labels, then test the model's zero-shot performance on UCF101 under different noise rate scenarios. We find that adversarial training, though promising under closed-set scenarios in previous studies, is relatively ineffective under open-vocabulary settings. Specifically, training with more noise lowers significantly the model's performance under low noise rate. Additionally, its added value is limited under heavy noise rate. These phenomena are probably related to the domain gap between datasets. By training on noisy text descriptions, the model tends to overfit the noise pattern, jeopardizing its zero-shot performance. We conclude that noisy text descriptions are better solved in testing time rather than during training stage. Our DENOISER framework shows a significant advantage over the adversarial training.

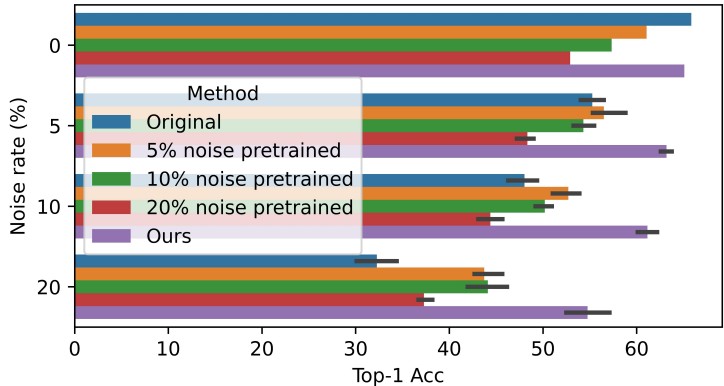

Figure 5: **Comparison to Adversarial Training.** Adversarial training is not efficient, especially in low-noise scenarios, even leading to a lower performance compared to the original model. It also falls behind our method by a significant margin.

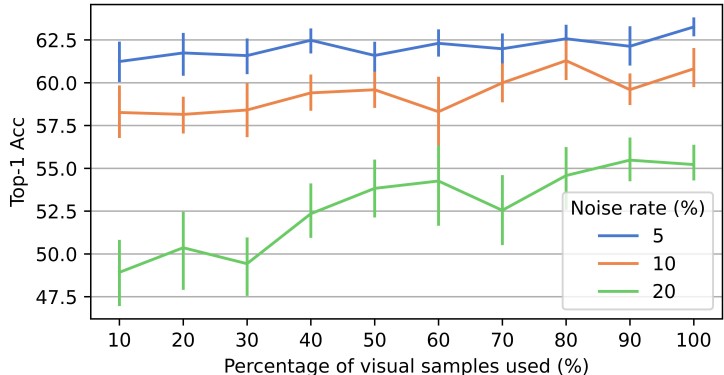

Figure 6: **Ablation Study on the Number of Visual Samples**. When fewer visual samples are used in $\Phi_{\text{inter}}$, our method shows a drop in performance. The bigger the noise rate, the larger the drop, showing that $\Phi_{\text{inter}}$ plays a role of increasing importance when the noise is larger.

## B.2 Ablation Study on the Number of Visual Samples

Fig. 6 ablates on the number of visual samples in $\Phi_{\text{inter}}$. Our method shows a drop in performance when fewer visual samples are used in $\Phi_{\text{inter}}$. The performance tends to converge towards that when solely $\Phi_{\text{intra}}$ is used. We hypothesize that fewer visual samples make $\Phi_{\text{inter}}$ harder to extract added value to $\Phi_{\text{intra}}$. With the noise rate increasing, we find an increasingly large drop in performance, which shows conversely that $\Phi_{\text{inter}}$ is more important under large noise scenarios as textual information becomes more ambiguous and less informative.

## B.3 Qualitative Results

Fig. 7 visualizes the embedding of (visual samples, text descriptions) from three semantic-classes: bird (green), ship (yellow), truck (blue) in CIFAR-10 using T-SNE. The first principal component of textual embedding is removed following ReCLIP[12] to prevent them from clustering at the same place. The Left shows that classification accuracy is low when text descriptions are noisy. Almost all visual samples are recognized as "bird". The Middle shows the embeddings of proposed text candidates. Some of them remain at the same place, because they move perpendicular to this 2D space in the real semantic space. We assign the best set of visual samples for each semantic-class to help denoise, *e.g.*, the blue dots are used to vote on the two candidates "trump" (red) and "truck" (purple) of "trumk". The Right shows that the denoised text descriptions improve the OVAR performance.

Tab. 4 quantifies some good/bad cases. We find GPT 3.5 is better at understanding semantics of noisy text descriptions, *e.g.*, "wal4ingm with a dog" → "dogwalking". However, its output is highly

Table 4: **Cases of Denoised Text Descriptions for GPT 3.5 and *DENOISER*.** The output from GPT 3.5 [1] tends to be unstable, and sometimes it's a relatively high-level understanding of noisy text descriptions. Our *DENOISER* ensures a relatively faithful output in terms of spelling but could be slightly mistaken when two words are similar in terms of both semantics and spelling.

|  | Ground Truth | Noisy Text Descriptions | GPT 3.5 [1] | **Ours** |
|---|---|---|---|---|
| Good Case | walking with a dog | wal4ingm with a dog | dogwalking | walking with a dog |
|  | baby crawling | babty crawling | baby crying | baby crawling |
|  | cutting in kitchen | cutting i_ aitnchen | cutting | cutting in kitchen |
| Bad Case | juggling balls | juggling ball_ | juggling | juggling ball_ |

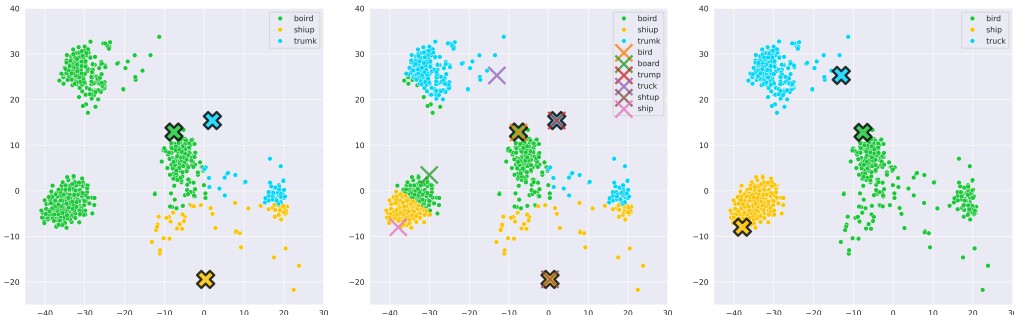

Figure 7: **Denoising Visualization. Left:** result with noisy text descriptions (crosses w black border). **Middle:** text candidates (crosses w/o black border), the visual samples (in dots) that are used to vote for candidates. **Right:** denoised class texts (crosses w black border) help for better classification.

affected by input prompts, and thus tends to be unstable: important text parts are sometimes omitted or misinterpreted, *e.g.*, "babty crawling" → "baby crying". Such unstable outputs require manual cleaning, limiting its applications in reality. Our *DENOISER* remains faithful in terms of spelling, *e.g.*, "wal4ingm with a dog" → "walking with a dog" instead of "dogwalking". While it may be mistaken when two words are similar in semantics and spelling (rare cases), *e.g.*, "ball" and "balls".

## C  On the efficiency of DENOISER

Our model requires a trade-off between computational cost and performance. As shown in Fig. 4 and Fig. 6, the performance of our *DENOISER* increases as the number of proposals $K$ and the percentage of the visual samples used. Since the theoretical complexity of *DENOISER* increases linearly with $K$ and the percentage of visual samples used, while the marginal contribution of a larger $K$ or percentage is decreasing, a trade-off between computational cost and performance is necessary.

*DENOISER* requires only simple operations for each iteration. After having extracted the embedding of visual samples, *DENOISER* only requires recomputing the text embedding and doing a dot product with visual embeddings, which is extremely fast. Compared to other approaches that intend to align noisy text-image pairs or to train spell-checking models, *DENOISER* that denoises at evaluation time is extremely time-saving.

