# OpenReview forum: "DENOISER: Rethinking the Robustness for Open-Vocabulary Action Recognition"
_NeurIPS.cc/2024/Conference — Submitted to NeurIPS 2024_

### Official Review · Reviewer_bX9i · 2024-07-10

**Soundness:** 3
**Presentation:** 2
**Contribution:** 2
**Rating:** 5
**Confidence:** 4

**Summary:**

This paper proposes a denoising framework to alleviate the influence of noisy text descriptions on open-vocabulary action recognition in real scenarios. A comprehensive analysis of the noise rate/type in text description is provided and the robustness evaluation of existing OVAR methods is conducted. A DENOISER framework with generative-discriminative optimization is proposed. The experiments demonstrate the effectiveness of the framework.

**Strengths:**

- The robustness to noisy text descriptions/instructions in real-world OVAR applications is an interesting and meaningful problem.
- The evaluation of the robustness of existing OVAR methods when facing the noise text description input is valuable to the community.
- The motivation is clear and the overall framework is technically sound.

**Weaknesses:**

- About the experiments,
    - The reviewer thinks that the most convincing results are the Top-1 Acc of existing OVAR models under the Real noise type. However, in Table 1, the proposed model does not demonstrate much superiority compared to GPT3.5's simple correction. The reviewer worries about the research significance of this problem. Will this problem be well resolved when using more powerful GPT4/GPT4o with some engineering prompt designs?
    - In Table 2, I would like to see the performance of other correction methods (e.g., GPT3.5/4/4o) for a more comprehensive comparison.
    - Since this work focuses on the noise text description problem in OVAR, it is necessary to demonstrate the results of those CLIP-based methods without any additional textual adaptation (e.g., the vanilla CLIP).


- About the method,
    - The reviewer thinks that the overall model design is reasonable and clear. However, the method part introduces too many symbols which makes the paper very hard to follow. It is unnecessary to over-decorate the technical contributions.

- Minor issue,
    - The authors seem to have a misunderstanding about the OVAR setting (Line 113). In OVAR, the model is evaluated on both base-classes and novel-classes during testing. In this case, all action classes from the UCF/HMDB datasets can be used for testing when the model is trained on K400, as there are many overlap classes between K400 and UCF/HMDB.

**Questions:**

Please see the weaknesses section.

**Limitations:**

The limitations are discussed and there is no potential negative societal impact.

---

> ### Author Rebuttal · Authors · 2024-08-07
>
> **W1** In Table 2, I would like to see the performance of other correction methods (e.g., GPT3.5/4/4o) for a more comprehensive comparison.
>
> **A1** We present additional results tested on UCF101 with ActionCLIP, using GPT4o and GPT4 as denoising model. We will update table 2 with result of GPT4/4o in the final version. We did each experiment 5 times.
> |        | 5% | 10% |
> |:------:|:--:|:---:|
> | GPT3.5 |  59.7±1.2 | 58.5±1.3  |
> |  GPT4o |  58.0±2.4 |  56.6±0.5  |
> |  GPT4  |  60.6±1.8 |58.7±1.0  |
> |  Ours  |  **63.8±0.7** | **61.2±0.8**  |
>
> Admittedly, GPT is a respected competitor and shows significant improvement over other unimodal text denoising models. However, the training and inference costs of GPT are much higher than our method. With our lightweight design that leverages both textual and visual information, we have achieved higher performance than GPT.
>
> Furthermore, in the text proposal module, we use only a corpus-based approach. We anticipate that incorporating a more powerful generative language model, such as GPT, has the potential to further enhance the performance of our method. It could be an interesting point for future research.
>
> **W2** Since this work focuses on the noise text description problem in OVAR, it is necessary to demonstrate the results of those CLIP-based methods without any additional textual adaptation (e.g., the vanilla CLIP).
>
> **A2** We acknowledge that most of the OVAR models use prompt engineering and totally agree that it is necessary to compare the performance without prompt. We test ActionCLIP on UCF101:
>
> |   ActionCLIP  | w/o ours | w ours |
> |:---:|:--------:|:------:|
> |  0% |   66.3    | /  |
> |  5% |  54.9±1.8    | 63.2±0.7  |
> | 10% | 47.3±1.4   | 61.2±1.2 |
>
> |   ActionCLIP  without prompt  | w/o ours | w ours |
> |:---:|:--------:|:------:|
> |  0% |   66.4   | /  |
> |  5% |  55.0±2.1    | 63.0±1.1  |
> | 10% | 47.0±3.2   | 61.0±1.3 |
>
> We find that prompt engineering does not seem to improve the robustness of OVAR model nor affect the effectiveness of our method.
>
> **W3** The reviewer thinks that the overall model design is reasonable and clear. However, the method part introduces too many symbols which makes the paper very hard to follow. It is unnecessary to over-decorate the technical contributions.
>
> **A3** We would like to thank the reviewer for this advice. We will sincerely consider every possibility to simplify the theoretical derivation while keeping the main ideas intact in the final version.
>
> **W4** The authors seem to have a misunderstanding about the OVAR setting (Line 113). In OVAR, the model is evaluated on both base-classes and novel-classes during testing. In this case, all action classes from the UCF/HMDB datasets can be used for testing when the model is trained on K400, as there are many overlap classes between K400 and UCF/HMDB.
>
> **A4** We would like to thank the reviewer for pointing it out. We will reformulate the definition of OVAR in a more rigorous way in the final version.

---

> > ### Author Response · Authors · 2024-08-11
> >
> > Dear Reviewer bX9i,
> >
> > We hope this message finds you well. I am reaching out to kindly request your prompt response to confirm whether our responses adequately address your queries. We sincerely thank you for your time and effort during this discussion period. Your timely feedback is greatly appreciated.

---

> > ### Comment · Reviewer_bX9i · 2024-08-11
> > **post-rebuttal-1**
> >
> > Thanks to the authors for their efforts during the rebuttal. After carefully reading the responses, my concerns are well resolved. Considering the concerns from other reviewers, I decided to keep my rating at "borderline accept".

---

> ### Author Response · Authors · 2024-08-12
>
> Thanks Reviewer bX9i for the timely reply.
>
> We are pleased to see that our rebuttal has addressed all the reviewer's concerns.
>
> If any remaining concerns hold the reviewer's opinion on the current borderline recommendation, we would be happy to provide further clarification and discussion.
>
> Since the reviewer feels that we have proposed an interesting and meaningful problem and that our contribution is valuable to the community. We invite the reviewer to kindly raise the recommendation rating. We can't let an interesting and worthwhile endeavor go unnoticed.

---

### Official Review · Reviewer_5bE7 · 2024-07-10

**Soundness:** 2
**Presentation:** 3
**Contribution:** 2
**Rating:** 4
**Confidence:** 5

**Summary:**

This paper tackles the challenge of noisy text descriptions in Open-Vocabulary Action Recognition (OVAR), a task that associates videos with textual labels in computer vision. The authors identify the issue of text noise, such as typos and misspellings, which can hinder the performance of OVAR systems. To address this, they propose a novel framework named DENOISER, which consists of generative and discriminative components. The generative part corrects the noisy text, while the discriminative part matches visual samples with the cleaned text. The framework is optimized through alternating iterations between the two components, leading to improved recognition accuracy and noise reduction. Experiments show that DENOISER outperforms existing methods, confirming its effectiveness in enhancing OVAR robustness against textual noise.

**Strengths:**

- This paper aims to study a new research topic, i.e., achieving robust open-vocabulary recognition performance with noisy texts. This direction has not been investigated before, which seems to be applicable in real-world applications.

- The proposed intra-modal and inter-modal methods are intuitive and demonstrated effective in the experiments.

- The experiments show that the proposed method is effective with different network architectures (XCLIP and ActionCLIP), which verifies that the method can be widely used.

**Weaknesses:**

- The baseline models are outdated and not tailored for OVAR. The authors failed to reference recent OVAR papers such as OpenVCLIP[1] (ICML 2023), FROSTER (ICLR 2024), and OTI (ACM MM 2023).

- In Table 1, it is evident that the proposed method outperforms GPT-3.5. Additionally, the authors present examples in Table 4 to demonstrate the superiority of the proposed method over GPT-3.5. However, upon personal experimentation with all the examples from Table 4 using the provided prompt from the paper (lines 243-245), I observed that the GPT-3.5 model successfully rectified all issues, including challenging cases where the proposed method fell short. As a result, I remain unconvinced by the findings.

This is the prompt given to GPT-3.5 model, and I hope other reviewers can also try it on their own:

The following words may contain spelling errors by deleting, inserting, and substituting letters. You are a corrector of spelling errors. Give only the answer without explication. What is the correct spelling of the action of  “cutting i aitnchen”.


[1] Open-VCLIP: Transforming CLIP to an Open-vocabulary Video Model via Interpolated Weight Optimization.

[2] FROSTER: Frozen CLIP Is A Strong Teacher for Open-Vocabulary Action Recognition.

[3] Orthogonal Temporal Interpolation for Zero-Shot Video Recognition.

**Questions:**

Please refer to the weaknesses.

**Limitations:**

Yes, they addressed the limitations.

---

> ### Author Rebuttal · Authors · 2024-08-07
>
> **W1** The baseline models are outdated and not tailored for OVAR. The authors failed to reference recent OVAR papers such as OpenVCLIP[1] (ICML 2023), FROSTER (ICLR 2024), and OTI (ACM MM 2023).
>
> **A1**
> Please note that in this paper, we are motivated is to draw public attention to the poor robustness of OVAR models against noisy textual descriptions, and to make the first attempt to mitigate this issue. Pursuing SOTA on the OVA task is not our primary intention.
>
> ActionCLIP and X-CLIP are two classic models, which are widely-adopted by the community, in the OVAR task (They are widely adopted as benchmark in recent paper such as OpenVCLIP、FROSTER、OTI). We believe that testing on these models have wider real-world implications and are convinced that they are representative enough to demonstrate the poor robustness of OVAR models and the generalizability of our method.
>
> To address concerns about the effectiveness of DENOISER on recent OVAR models, we provide additional results using Open-VLIP as one instance:
> |   ActionCLIP  | w/o ours | w ours |
> |:---:|:--------:|:------:|
> |  5% |  54.9±1.8    | 63.2±0.7  |
> | 10% | 47.3±1.4   | 61.2±1.2 |
>
> |   X-CLIP  | w/o ours | w ours |
> |:---:|:--------:|:------:|
> |  5% |    55.6±2.2  |  64.2±1.4 |
> | 10% |  46.4±1.3  |  62.9±2.3 |
>
> |Open-VLIP| w/o ours | w ours |
> |:---:|:--------:|:------:|
> |  5% |   61.8±1.3  |  67.8±1.0 |
> | 10% |   47.6±2.3  |  64.7±3.0 |
>
>
> We find that SOTA OVAR models, such as Open-VLIP, are also susceptible to noise. Moreover, our method is effective on these models as well. Furthermore, we validate our finding (see Lines 250-256 of the main paper) that the better the underlying model, the better the final outcome will be with our method, demonstrating that our model is scalable.
>
> We will do more validation on these recent models such as FOSTER and OTI and cite them in the final version.
>
> **W2** In Table 1, it is evident that the proposed method outperforms GPT-3.5. Additionally, the authors present examples in Table 4 to demonstrate the superiority of the proposed method over GPT-3.5. However, upon personal experimentation with all the examples from Table 4 using the provided prompt from the paper (lines 243-245), I observed that the GPT-3.5 model successfully rectified all issues, including challenging cases where the proposed method fell short. As a result, I remain unconvinced by the findings.
>
> **A2** We would like to thank the reviewer for acknowledging the superiority of our proposed method over GPT-3.5. We understand that GPT’s output can be variable, which is why we conduct each experiment 10 times and report the confidence intervals. We provide a script to query GPT-3.5, which we use to obtain the denoised examples for performance evaluation. We encourage other reviewers to try it as well.  We will include logs in the code release to substantiate our findings.
>
> We feel that coming a conclusion that GPT can rectify all issues is arbitrary and far from rigorous. In tables 4, we present merely some of the typical examples that we have encountered in our experiments. To be honest, we queried GPT-3.5 with the exact prompt:
>
> <<The following words may contain spelling errors by deleting, inserting, and substituting letters. You are a corrector of spelling errors. Give only the answer without explication. What is the correct spelling of the action of “cutting i aitnchen”. >>
>
> Here is the 10 outputs of GPT: cutting a kitchen", "cutting in aitch", "cutting inaction", "cutting in action", "cutting in aitching", "cutting itchin", "cutting a kitchen", "cutting a kitchen", "cutting in action", "cutting in action".

---

> > ### Author Response · Authors · 2024-08-11
> >
> > Dear Reviewer 5bE7,
> >
> > We hope this message finds you well. I am reaching out to kindly request your prompt response to confirm whether our responses adequately address your queries. We sincerely thank you for your time and effort during this discussion period. Your timely feedback is greatly appreciated.

---

> ### Comment · Reviewer_5bE7 · 2024-08-12
> **Response to Rebuttal**
>
> Thanks to the authors for their detailed rebuttal.
>
> Regarding the first point, I recommend the authors to include additional experiments and references in the revision.
>
> Regarding the second point, I tried the prompt provided by the authors with the GPT-3.5 model ten times, and I found the model can correct the spelling for every run.
>
> Below is the log:
>
> 1. The correct spelling is "cutting in the kitchen".
> 2. The correct spelling is "cutting in the kitchen."
> 3. The correct spelling is "cutting in the kitchen".
> 4. The correct spelling is "cutting in the kitchen".
> 5. The correct spelling is "cutting in a kitchen".
> 6. The correct spelling is "cutting in a kitchen."
> 7. The correct spelling is "cutting in a kitchen."
> 8. The correct spelling is "cutting in a kitchen".
> 9. The correct spelling is "cutting in the kitchen".
> 10. The correct spelling is "cutting in the kitchen."
>
> Therefore, I am still not convinced by the rebuttal.
>
> After consideration, I decide to raise my score to 4.

---

> > ### Author Response · Authors · 2024-08-12
> >
> > Thanks Reviewer 5bE7 for your timely reply.
> >
> > We feel that we cannot agree with each other on the specific example of "cutting i aitnchen". Thus, we propose that we could try another one.
> >
> > Here is the setting for OpenAI API:
> > - Model: "gpt-3.5-turbo"
> > - System Message: "The following words may contain spelling errors by deleting, inserting and substituting letters. You are a corrector of spelling errors. Give only the answer without explication."
> > - User Message: "What is the correct spelling of the action of "writing on boarUd"."
> >
> > Here is the output of GPT:
> > - whiteboard
> > - board
> > - board
> > - whiteboarding
> > - board
> > - board
> > - board
> > - board
> > - board
> > - board
> > - board
> > - The correct spelling is "writing on board".
> > - board
> > - board
> > - board
> > - board
> > - whiteboarding
> > - board
> > - board
> > - "boarding"
> > - board
> > - board
> > - onboard
> >
> > Furthermore, we believe that focusing on a single example may lead us to ignore the whole picture. Instead of focusing on a single example, we would like to present here how GPT and our method denoise the text descriptions in a quantitative way.
> >
> > Experiments conducted under a noise rate of 5% on UCF101:
> > |        | Acc-0 | Acc-1 | Acc-2 | Mean Edit Distance |
> > |:------:|:-----:|:-----:|:-----:|:----------:|
> > | GPT3.5 | 47.63 | 71.49  |  73.77 |   2.59   |
> > |  Ours  | 65.69  |  80.70  | 91.75 |   0.67   |
> > where:
> > - Edit Distance : Levenshtein edit distance
> > - Acc-0: Denoised text description matches exactly the original text description
> > - Acc-1: Denoised text description is of edit distance<=1 from the original one
> > - Acc-2: Denoised text description is of edit distance<=2 from the original one
> > - Mean Edit Distance: mean edit distance between the denoised text description and the original one
> >
> > GPT may recover on average 73% of the time, a denoised text who has an edit distance smaller than 2 with the original text, which is still consistent with the reviewer's experiment.
> >
> > We are not trying to deny the power of GPT in such tasks. GPT is a respected competitor, yet not tailed for such tasks.
> >
> > Our method surpasses GPT by a notable margin, without a complex language model.

---

> > > ### Comment · Reviewer_5bE7 · 2024-08-12
> > > **Response to Comments**
> > >
> > > Thanks for your comment.
> > >
> > > Could you please provide me the code that you query GPT-3.5 for correcting the issues? And, it would be better you provide the cases that you find GPT fails to solve while your method successfully corrects.

---

> > > > ### Author Response · Authors · 2024-08-12
> > > >
> > > > ```python
> > > > import json
> > > > import http.client
> > > >
> > > > conn = http.client.HTTPSConnection("api.openai.com")
> > > > payload = json.dumps({
> > > > "model": "gpt-3.5-turbo",
> > > > "messages": [
> > > >         {
> > > >             "role": "system",
> > > >             "content": 'The following words may contain spelling errors by deleting, inserting and substituting letters. You are a corrector of spelling errors. Give only the answer without explication.',
> > > >         },
> > > >         {
> > > >             "role": "user",
> > > >             "content": f'What is the correct spelling of the action of "writing on boarUd".'
> > > >         }
> > > > ],
> > > > })
> > > > headers = {
> > > > 'Accept': 'application/json',
> > > > 'Authorization': 'Bearer your api key',
> > > > 'Content-Type': 'application/json'
> > > > }
> > > > conn.request("POST", "/v1/chat/completions", payload, headers)
> > > > res = conn.getresponse().read().decode("utf-8")
> > > > res = json.loads(res)['choices'][0]['message']['content']
> > > > ```

---

> > > > > ### Comment · Reviewer_5bE7 · 2024-08-12
> > > > > **Response to Authors**
> > > > >
> > > > > Thank you for sharing your code.
> > > > >
> > > > > Using the API, I was able to successfully reproduce your results. Interestingly, when I tested the cases directly on the GPT-3.5 conversation webpage, I found that the model could resolve all the issues. This suggests there may be some underlying factors at play. Regardless, I am now convinced by the results.
> > > > >
> > > > > After reviewing the feedback from Reviewer eAxG, I agree that this method has no specific designs for OVAR and should be tested on a broader range of tasks. A more comprehensive validation would undoubtedly make the paper more convincing.
> > > > >
> > > > > Therefore, I believe the paper needs additional experiments across more benchmarks, such as image-based tasks, to be considered ready for a conference like NIPS. I appreciate the efforts the authors have put into the rebuttal and discussion. If they address the reviews and improve the paper accordingly, it will become a more compelling piece of work.
> > > > >
> > > > > Considering all the discussion, I keep my score to be borderline reject.
> > > > >
> > > > > Thank you.

---

### Official Review · Reviewer_eAxG · 2024-07-13

**Soundness:** 3
**Presentation:** 3
**Contribution:** 2
**Rating:** 3
**Confidence:** 5

**Summary:**

This paper deals with the problem of Open-Vocabulary Action Recogniton (OVAR). Specifically, it focuses on the issue that the action labels provided by users may contain some noise such as misspellings and typos. The authors find that the existing OVAR methods' performance drops significantly in this situation.  Based on this analysis, they propose the DENOISER framework to reduce the noise in the action vocabulary.

**Strengths:**

1. The paper is generally well-written and easy to follow.
2. The framework is well presented and explained.
3. The experiments show the effectiveness of the denoising process.

**Weaknesses:**

1. This paper actually focuses on text denoising and does not involve any specific action recognition technology. The author just chose the field of OVAR to verify the effectiveness of the proposed text-denoising method. The title is somewhat misleading. I think the author should regard text-denoising as the core contribution of the article instead of the so-called "robust OVAR"
2. The article focuses on too few and too simple types of text noise, including only single-letter deletions, insertions, or substitutions. These kinds of errors can be easily discovered and corrected through the editor's automatic spell check when users create a class vocabulary. This makes the method in this paper very limited in practical significance.
3. , The proposed method, although a somewhat complex theoretical derivation is carried out in the article, is very simple and intuitive: that is, for each word in the class label, selecting the one that can give the highest score to the sample classified into this category among several words that are closest to the word.  There is limited novelty or technical contribution.

**Questions:**

See weakness.

**Limitations:**

The author states two limitations of the work in the paper.

---

> ### Author Rebuttal · Authors · 2024-08-07
>
> **W1**
> This paper actually focuses on text denoising and does not involve any specific action recognition technology. The author just chose the field of OVAR to verify the effectiveness of the proposed text-denoising method. The title is somewhat misleading. I think the author should regard text-denoising as the core contribution of the article instead of the so-called "robust OVAR"
>
> **A1** We underline that this paper does not focus only on denoising texts. Its contribution is two-fold:
> - *Robustness Verification*: We verify the robustness of existing OVAR models under noisy textual descriptions, highlighting the poor robustness of such models to the community.
> - *Robustness Improvement*: We make the first attempt to improve the robustness of OVAR models through text denoising.
> Considering these two contributions, we believe it is reasonable to regard robust OVAR as the core contribution of this paper. Furthermore, we agree with the reviewer that the proposed method has the potential to generalize to other domains involving visual-textual alignment. We hope to see more research investigating the robustness of models in these domains in the future.
>
> **W2** The article focuses on too few and too simple types of text noise, including only single-letter deletions, insertions, or substitutions. These kinds of errors can be easily discovered and corrected through the editor's automatic spell check when users create a class vocabulary. This makes the method in this paper very limited in practical significance.
>
> **A2** Following a large amount of previous literature [1,2,3], there are three main types of textual noise: insertion, deletion, and substitution. Real-world scenarios typically involve a mix of these types. We hence carefully determined the noise rate using real-world corpora [4,5].
>
> These types of errors, although seemingly intuitive, are proven to be difficult for naive spell-checking models or even GPT to denoise. In this paper, we demonstrate that under a 10% noise rate, using ActionCLIP as the OVAR model to test on UCF101, a vocabulary-based method (PySpellChecker) achieves only 55.7% Top-1 accuracy, which is 5.5% behind our method; GPT-3.5 achieves 58.5%, which is 2.7% behind our method. This further shows that our noise setting is not as simple as one might imagine. On the contrary, it is complicated enough for practical scenarios.
>
> As the first paper to rethink the robustness of OVAR models under noise, we believe our settings are practical enough to reveal the poor robustness of these models and to inspire future work in this domain.
>
> [1] Spelling Errors in English Writing Committed by English-Major Students at BAU
>
> [2] Models in the Wild: On Corruption Robustness of Neural NLP Systems
>
> [3] Benchmarking Robustness of Text-Image Composed Retrieval
>
> [4] A Benchmark Corpus of English Misspellings and a Minimally-supervised Model for Spelling Correction
>
> [5] GitHub Typo Corpus: A Large-Scale Multilingual Dataset of Misspellings and Grammatical Errors
>
>
> **W3** The proposed method, although a somewhat complex theoretical derivation is carried out in the article, is very simple and intuitive: that is, for each word in the class label, selecting the one that can give the highest score to the sample classified into this category among several words that are closest to the word. There is limited novelty or technical contribution.
>
> **A3** Thank you for acknowledging the intuitiveness of our method. As a first attempt to tackle the robustness of OVAR models, our simple and intuitive approach achieves remarkable performance, even compared to GPT. Furthermore, the idea of a generative-discriminative joint optimization framework behind DENOISER, backed by theoretical derivation, has great potential beyond this paper. For example, we could leverage large language models (LLMs) to better model the intra-modal weight while using state-of-the-art multi-modal models to enhance the inter-modal weight. Thus, we believe that this paper does bring non-trivial contributions to the community,

---

> > ### Author Response · Authors · 2024-08-11
> >
> > Dear Reviewer eAxG,
> >
> > We hope this message finds you well. I am reaching out to kindly request your prompt response to confirm whether our responses adequately address your queries. We sincerely thank you for your time and effort during this discussion period. Your timely feedback is greatly appreciated.

---

> > > ### Comment · Reviewer_eAxG · 2024-08-12
> > >
> > > Thanks to the authors for their efforts during the rebuttal. I have read all the comments from other reviewers and the authors' responses. I agree with Reviewer 5bE7 that GPT's performance in text correction may be underestimated and agree with Reviewer bX9i that the technical contribution of the proposed method is somewhat over-decorated. Besides, I stand by my opinion that this paper actually focuses on text denoising and does not involve any specific action recognition technology. The proposed text-denoising method should be evaluated on more text-related tasks beyond OVAR. The types of text noise that the paper focuses on are too few and too simple. For OVAR, considering that the largest action recognition dataset has only hundreds of categories, even if there is a 10% error rate, the errors can be easily discovered through the editor's automatic spell check and then corrected manually when users create a class vocabulary. Only in fields where the amount of text data is very large, the text denoising method proposed will have practical significance. I decide to keep my initial rating for the above reasons.

---

### Official Review · Reviewer_YJ4Z · 2024-07-13

**Soundness:** 2
**Presentation:** 3
**Contribution:** 2
**Rating:** 4
**Confidence:** 3

**Summary:**

This paper addresses the challenge of noisy text descriptions in  Open-Vocabulary Action Recognition. It introduces the DENOISER  framework, which combines generative and discriminative approaches to denoise the text descriptions and improve the accuracy of visual sample  classification. The paper provides empirical evidence of the framework's  robustness and conducts detailed analyses of its components.

**Strengths:**

1. The paper is well-written and the content is easy to understand.
2. The motivation presented by the authors is clear, the label noise problem does exist in video datasets.
3. The authors show the types of noise and their percentage, in addition, the authors verify the validity of the proposed method through comparative experiments.

**Weaknesses:**

1. As the authors state in the limitations section, textual description noise does exist, but it can be corrected with an offline language model, what are the advantages of the authors' proposed approach?
2. I would assume that the text noise problem presented in this paper is even worse on large video datasets collected by semi-automatically labeled networks, e.g., Panda70M, howto100M, and InternVid. I suggest that the authors might consider validating their ideas on these datasets.

**Questions:**

1. Citation [49] Figure 3, reports ActionClip's zeroshot Top-1 accuracy performance for HMDB-51 and UCF-101, which is 50% and 70%, respectively. Why is it different from the baseline results in Table 2 in this paper?
2. Some contrastive learning pre-training methods for text noise have been proposed, e.g., [1-2], and I have not seen any relevant discussions or experimental comparisons in the papers.

[1] Karim, Nazmul, et al. "Unicon: Combating label noise through uniform selection and contrastive learning." Proceedings of the IEEE/CVF conference on computer vision and pattern recognition. 2022.
[2] Ghosh, Aritra, and Andrew Lan. "Contrastive learning improves model robustness under label noise." Proceedings of the IEEE/CVF Conference on Computer Vision and Pattern Recognition. 2021.

**Limitations:**

The authors have provided a limitations analysis in their paper and I have suggested some limitations in the Questions section.

---

> ### Author Rebuttal · Authors · 2024-08-07
>
> **W1** Textual description noise does exist, but it can be corrected with offline language model, what are the advantages of the proposed approach?
>
> **A1**  Thank you for acknowledging the presence of noise in text descriptions. We agree that offline language models can mitigate these noises to an extent. However, a unimodal approach has two drawbacks (please see lines 48-54 of our paper):
>
> - *Textual Ambiguity*: For example, the noisy text "boird" could be equally interpreted as "bird" or "board." Visual information can help the model better understand the context.
>
> - *Cascaded Errors*: When text denoising and action recognition are performed independently without shared knowledge, errors from text correction are propagated to action recognition, resulting in continuous error propagation.
>
> To address these issues, we propose the cross-modal denoising model DENOISER. This model not only retains the benefits of unimodal models through intra-modal weighting but also leverages visual information through inter-modal weighting. By alternating generative-discriminative steps, we overcome the aforementioned problems:
>
> - *Ambiguity Resolution*: Cross-modal information helps clarify textual ambiguities.
>
> - *Error Propagation Prevention*: Joint optimization of generative and discriminative steps couples text denoising and action recognition, thus avoiding cascaded error propagation.
>
> Experimentally, we demonstrate the superiority of our method compared to traditional language models (e.g., BERT and PySpellChecker). Even when compared to GPT-3.5, our method yields a 4% improvement. Here is Top1 Acc of ActionCLIP on UCF101 under 5% and 10% noise rate:
> ||5%|10%|
> |:-:|:-:|:-:|
> |GPT3.5|59.7±1.2|58.5±1.3|
> |Bert(NeuSpell)|56.6±0.5|51.0±0.5|
> |PySpellChecker|60.9±1.1|55.7±1.1|
> |Ours|**63.8±0.7**|**61.2±0.8** |
>
> **W2** Text noise problem is even worse on large video datasets collected by semi-automatically labeled networks, e.g., Panda70M, howto100M, and InternVid. I suggest that the authors might consider validating their ideas on these datasets.
>
> **A2** Thank you for this valuable proposal. We agree that large video datasets may contain additional noise. However, they present several challenges:
>
> - *Varied Noise Types*: These datasets include a wide range of noise types, such as textual noise, image-text misalignments, and image distortions. This diversity complicates the direct validation of our core concepts: how to mitigate noise in text descriptions for OVAR task.
>
> - *Benchmarking Issues*: These datasets are not standard benchmarks in the OVAR domain, making it difficult to perform fair comparisons with existing methods.
>
> To ensure fair comparisons with other models, we validate our method on K700, K400, UCF101, and HMDB51 (K700 and K400 are also large in scale), which encompass nearly all major action recognition benchmarks， ensuring that our conclusions are generalizable and credible.
>
> **Q1** Citation [49] Figure 3, reports ActionClip's zeroshot Top-1 accuracy performance for HMDB-51 and UCF-101, which is 50% and 70%, respectively. Why is it different from the baseline results in Table 2 in this paper?
>
> **A3** Sorry for the confusion.
>
> For HMDB51, we directly used the open-source code and model released on GitHub to evaluate zero-shot performance of ActionCLIP, which achieved 46.16% Top-1 accuracy. Additionally, we refer the reviewer to page 11, Table 3 of the X-CLIP paper [1], who reports zero-shot performance of ActionCLIP on HMDB51 as 40.8 ± 5.4%. This result is consistent with our experiments.
>
> For UCF101, to maintain consistency with other datasets and settings of other models, we lowercase the labels and add spaces between words. Note that in HMDB51 and Kinetics datasets, labels naturally include spaces. Open-VLIP also uses UCF101 labels with spaces (its performance drops from 83.4% to 76.7% with the original UCF101 labels without spaces). This difference in performance further demonstrates the poor robustness of these models. We will provide the log in code release to substantiate our finding.
>
> [1] Expanding Language-Image Pretrained Models for General Video Recognition
>
> **Q2** Some contrastive learning pre-training methods for text noise have been proposed, e.g., [1-2], and I have not seen any relevant discussions or experimental comparisons in the papers.
>
> **A4** The reviewer may have some misunderstandings of our paper.
>
> A typical OVAR task usually consists of two steps:
> - Step 1: Pretrain or Fine-tune a visual-textual alignment model
> - Step 2: Infer with encoded text descriptions and visual information
>
> Text noise exists in both steps. Papers mentioned by the reviewer focus on pretraining a robust visual-textual alignment model to handle noise in step 1. In contrast, this paper focuses on mitigating noise in step 2.
>
> Admittedly, we considered whether pretraining models with text noise in step 1 would enhance robustness in step 2. We present the performance of ActionCLIP finetuned with noisy texts on K400 (also see supplementary materials for more discussions). Each row shows the performance tested on UCF101 with no noise, 5% noise, and 10% noise in text descriptions.
> ||Original|5% noise finetune|10% noise finetune|Ours|
> |:-:|:-:|:-:|:-:|:-:|
> |0%|65.8|61.1|57.3|**65.1**|
> |5%|55.3±1.4|56.5±1.8|54.3±1.2|**63.2±0.7**|
> |10%|48.0±1.9|52.7±1.3|50.2±1.0|**61.2±1.2**|
>
> We found that pretraining on noisy text descriptions slightly improves robustness of original model. Nevertheless, it not only falls behind our method but also harms performance on clean samples. Models trained on clean samples are not robust against noise during testing period, while models trained on noisy samples fail to perform well on clean samples. This dilemma must be faced when trying to robustify a model through costly pretraining.
>
> In contrast, our method uses only information encoded in the OVAR model. It is cost-efficient and outperforms pretraining-based methods on both clean and noisy samples.

---

> > ### Author Response · Authors · 2024-08-11
> >
> > Dear Reviewer YJ4Z,
> >
> > We hope this message finds you well. I am reaching out to kindly request your prompt response to confirm whether our responses adequately address your queries. We sincerely thank you for your time and effort during this discussion period. Your timely feedback is greatly appreciated.

---

> > > ### Author Response · Authors · 2024-08-14
> > >
> > > Dear Reviewer YJ4Z,
> > >
> > > We have responded to your comments in detail. As the discussion period will end in less than 5 hours, we would like to ask whether there are any additional concerns or questions we could address.
> > >
> > > Thanks very much for your effort!
> > >
> > > Best regards,
> > >
> > > Authors

---

### Author Rebuttal · Authors · 2024-08-07

We thank the reviewers for their valuable advices.

We would like to underline that the contribution of this paper is two fold.

First, we verify the robustness of existing OVAR models during the inference period under noisy textual descriptions. Our study spans from classic models like ActionCLIP and XCLIP to more recent ones such as Open-VLIP, highlighting their poor robustness to the community.

Secondly, we make the first attempt to improve the robustness of OVAR models through text denoising. By leveraging both intra-modal and cross-modal information, our method outperfom not only traditional denoising language models (Pyspellchecker, Bert), but also GPT 3.5/4/4o.

We sincerely believe this paper is a valuable contribution to the community and earnestly hope it inspires further research into the robustness of OVAR models.

---

### Decision · Program_Chairs · 2024-09-25

**Decision:**

Reject

**Comment:**

This paper receives mixed reviews: three (borderline) rejects and one borderline accept. Some major concerns on the motivation of the proposed method, outdated baseline choice, and some unclear details. The AC agrees that the proposed technique should be more general with text denoising, which should not be only limited to open vocabulary action recognition. Therefore, the authors are encouraged to verify the effectiveness of proposed method in a more general setting. So, the AC thinks this paper is not ready for publication and makes a reject recommendation.